# Factors Influencing Self-Management Behaviors Among Patients with Post-Kidney Transplantation: A Qualitative Study of the Chronic Phase Transition

**DOI:** 10.3390/healthcare12222264

**Published:** 2024-11-13

**Authors:** Naoko Matsumura, Mariko Mizukawa, Kanae Sato, Asuka Hashino, Kana Kazawa, Makiko Naka, K. A. T. M. Ehsanul Huq, Michiko Moriyama

**Affiliations:** 1Community Cooperation & Medical Support Office, Tokuyama Central Hospital, Yamaguchi 745-0822, Japan; naomaz29@gmail.com; 2Ichikan Nursing Development Center for Diversity, Kobe City College of Nursing, Kobe 651-2103, Japan; 3Department of Nursing, Yasuda Women’s University, Hiroshima 731-0153, Japan; sato-ka@yasuda-u.ac.jp; 4Graduate School of Biomedical and Health Sciences, Hiroshima University, Hiroshima 734-8553, Japan; hasuka@kumamoto-u.ac.jp (A.H.); kkazawa@okayama-u.ac.jp (K.K.); ehsandr2002@yahoo.com (K.A.T.M.E.H.); morimich@hiroshima-u.ac.jp (M.M.); 5Department of Nursing, Faculty of Life Sciences, Kumamoto University, Kumamoto 862-0976, Japan; 6Faculty of Health Sciences, Okayama University, Okayama 700-8558, Japan; 7Heart Failure Center, Hiroshima University Hospital, Hiroshima 734-0037, Japan; nakamaki556@gmail.com

**Keywords:** self-management behavior, chronic kidney disease, kidney transplantation, renal transplantation, self-efficacy

## Abstract

Background: Kidney transplantation is an effective treatment for patients with kidney failure. Despite the advances in technology, a certain number of patients still deteriorate due to improper management. The purpose of this study was to identify the promoting and inhibitory factors that influence recipients’ self-management behaviors after a kidney transplant. Methods: We enrolled participants who had kidney transplants for more than one year, aged ≥20 years from outpatient clinics in Japan. Face-to-face interviews were conducted between April and December 2016. Results: Nine participants were included in this study. By qualitative content analysis, 115 codes and 8 categories were extracted for the factors resulting in maintenance and the promotion of self-management behaviors; those were [attentiveness to changes in one’s own body], [good partnership with medical care providers], [past painful experiences], [establishment of lifestyle habits], [autonomy to protect one’s own body], [support from family and others], [gratitude for kidney donation], and [increased self-efficacy]. We also extracted three categories that inhibited self-management behavior: [fading threat of worsening disease], [shifting priorities], and [decreased motivation to control the disease]. Conclusions: The passage of time after transplant became a barrier to continue self-management. Providing knowledge about the importance of self-management can prevent the deterioration of kidney function over time after a transplant.

## 1. Introduction

Kidney transplantation is an effective treatment for patients in the terminal stages of chronic kidney disease (CKD). It has been on the rise worldwide in recent years [1]. Kidney transplant outcomes have improved over the years, both in the short- and long-term [2], largely due to advances in immunosuppressive agents [3] and perioperative management [4]. As a result, the transition of recipient mortality has shifted from rejection and kidney dysfunction related to kidney transplant surgery to malignant diseases, cardiac diseases, cerebrovascular diseases, and other causes [5]. To prevent the onset of chronic diseases such as cardiovascular disease (CVD), the significance of self-management has long been noted [2]. The principles of self-management have not been changed over time [6]. Self-management has a positive impact before and after the kidney transplant by self-monitoring comorbidities including blood pressure, blood glucose, etc. to decrease morbidity and mortality [7,8,9].

The clinical practice guideline provides recommendations for managing post-transplant complications related to immunosuppressive agents, infections, malignancies, and CVD [10]. After the transplant, approximately 76% of recipients have CKD stage 3T or higher. Moreover, recipients of CKD stages 4T and 5T have a 8% and 49% risk of losing their graft in the following year, respectively [11]. Therefore, it is necessary to prevent the development and severity of medical complications and protect the function of the transplanted kidney. The recipients should adhere to the immunosuppressive drugs, check the side effects, increase self-management behaviors including the daily monitoring of weight and blood pressure, visit a doctor if measurements deviate, engage in regular infection-control behavior and salt restriction, prevent dehydration by adequate fluid intake, and monitor signs of reduced kidney function [12]. These actions can prevent kidney function decline and the development of lifestyle-related chronic illnesses. Therefore, recipients need to continue self-management behaviors and maintain their quality of life.

Prior research has emphasized the importance of kidney transplant recipients’ self-management behaviors [13]. Vanhoof et al. pointed out the changes in recipients’ post-transplant social backgrounds and roles. After the transplant, the business of daily life and deviations from their routine may affect self-management behaviors [14]. Pinter et al. and Quinn et al. focused on the post-transplant experience of recipients according to the characteristics of each age group, from adolescents to the elderly [15,16]. The recipients’ background factors influenced a good prognosis who had undergone more than 25 years of kidney transplant including “maintaining a healthy lifestyle”, “social support”, and “participation in decision-making”. Furthermore, it has been shown that recipients who maintained better self-management behavior before the transplant had a high success rate and experienced fewer complications [17].

Studies have clarified the experiences of self-management kidney post-transplant recipients, particularly the factors that maintain, promote, and inhibit self-management behaviors. Factors known to inhibit continued self-management include a prolonged post-transplant period, the complexity of the required treatment, adjusting to a new health status, depression, younger age, social isolation, and cognitive decline [6,18]. However, it is still unclear what the factors and the duration of inhibition for self-management behaviors after a transplant are [19,20].

The purpose of this study was to identify the promoting and inhibiting factors that influence recipients’ self-management behaviors after a kidney transplant. The results of this study can provide valuable suggestions for the effective and ongoing support for post-kidney transplant recipients to prevent the deterioration of kidney function and other complications by improving the self-care capacity. The kidney transplant care system can improve recipients’ quality of life more effectively.

## 2. Materials and Methods

### 2.1. Study Design

Face-to-face, semi-structured interviews were conducted using qualitative content analysis from April to December 2016. This study was conducted by the consolidated criteria for Reporting Qualitative research (COREQ) guidelines.

### 2.2. Study Participants and Study Site

Participants included post-kidney transplant recipients who visited the outpatient clinics of three medical institutions in Chiba, Sagamihara, and Shimonoseki cities in Japan, met all the following eligibility criteria, and consented to participate in the study.

The inclusion criteria were as follows: (i) post-kidney transplant recipients aged ≥20 years, regardless of gender, residency status, or family status at the time of transplant, (ii) recipients who underwent kidney transplant at least one year prior, (iii) any conditions for kidney transplant, regardless of living kidney/donated kidney transplant, length of time since transplant, or underlying disease leading to kidney failure, (iv) recipients who were recuperating at home, and (v) there were no physical or cognitive functional problems determined by the physician conducting the interview.

The reason for selecting patients one-year post-transplant is that, after one year, the interval between regular outpatient visits is longer, and this might result in fewer evaluations, less advice from medical care providers, and more self-directed management. The frequency of outpatient visits in Japan is at least once a month for the first year after a kidney transplant; however, after one year, outpatient visits tend to decrease. The exclusion criterion was participants who had cognitive decline (a score of less than 20, according to classification by the Mini-Mental State Examination: MMSE) [21].

### 2.3. Recruitment Procedure and Data Collection

Applicants who met the inclusion criteria were referred to the researcher (first author) by the study site specialists and kidney transplant recipient coordinators (RTCs). After obtaining written informed consent, the researcher started the interview. Participants’ recruitment and data collection were terminated when the researcher judged that their perceptions, understanding, and issues regarding the theme had been captured, i.e., when we achieved the study objectives, reached the saturation point, and found no new factors that were determined to be extracted.

The first author who was a graduate student in a master’s program in Clinical Nurse Specialists’ course with 7 years of experience in kidney transplant nursing conducted the interview, and the last (supervising) author who is a professor of chronic disease nursing with extensive experience in interview surveys supervised the study activities and data collection procedures. All researchers have sound knowledge about kidney transplant nursing and were not familiar with the study participants. Therefore, there were no chances of selection bias, assumptions, or any kind of selected research themes. The interviews were conducted in a private room in their clinics where privacy was secured.

The interview guide was developed based on previous literature, and the content was refined by nursing experts known to the research team, and then pretested by those who had experience of kidney transplants. Before the semi-structured interview, a structured questionnaire was administered to ask about the participants’ attributes and the background related to their transplants to help analyze the data obtained from the interview.

#### 2.3.1. Structured Interview Content

The content of the structured interview was based on the participants’ basic information (age, sex, education level, occupation, marital status, cohabitants and their relationship, and current physical condition); kidney transplant information (post-transplant period, pre-transplant complications, and donor source [living kidney donation or deceased donor]); presence or absence of dialysis before kidney transplant; type and years of dialysis; underlying disease leading to kidney failure; serum creatinine level; estimated glomerular filtration rate [eGFR]; presence or absence of disease, as well as the name of any disease mentioned; and current medical treatment status (outpatient visit intervals, average outpatient waiting time, the time required for outpatient visits, outpatient visiting transportation means, and presence or absence of emergency outpatient consultation use).

#### 2.3.2. Semi-Structured Interview Content

Self-management behaviors during recovery after kidney transplant regarding the effects of self-monitoring, daily living management after transplant, and early management of complications after transplant were interviewed. Additionally, the researcher asked recipients questions about their post-transplant self-management behaviors, including the reasons they continued their self-management behaviors and what motivated them to engage more. The following interview guide was used:

[Interview Guide]

What did you specifically recognize as self-management behavior before and after the kidney transplant? Have there been any deviations from the self-management behaviors you understood before the transplant that you had to take after the transplant, or in the actual self-management behaviors you have performed after the transplant?Do you think you have engaged in self-management behavior so far? Describe your progress and changes in self-management behavior after transplanting.Please explain the reason for the changes in self-management behaviors after your transplant that you mentioned in Question 2. (Your criteria for judgment in terms of self-management.)If you think you can no longer continue self-management, why? What caused you to stop continuing your self-management behavior? What were the specific situations and changes in your feelings at that time? (Factors and background influencing self-management behaviors.)What were your actions, thoughts, and feelings when you lost control of your self-management behaviors?How did you cope with your actions, thoughts, and feelings when you were unable to engage in self-management behaviors? Alternatively, if you could not deal with it, why not?

### 2.4. Data Analysis

The data were analyzed using qualitative content analysis as developed by Berelson [22]. First, the contents of the transcript were carefully read, and the parts expressing the factors that affected self-management behavior were extracted in context units for each meaningful piece of content to form a minimum unit code. The extracted codes were then compared and grouped into two categories for each code with similar content: factors that maintained or promoted self-management behavior and those that inhibited it. The number of codes was counted in each group. Finally, each group was set as a subcategory, and a category was generated accordingly. MS Excel was used to manage textual data. Furthermore, participants were divided into two groups: those who had transplants for less than and more than or equal to five years. We set the cutoff value at five years as the meantime elapsed following transplant for the nine participants. Then, we analyzed the codes extracted from the narratives and their attributes regarding the factors generated.

### 2.5. Trustworthiness

This study used the following four criteria to ensure the rigor of the qualitative research data, referring to Lincoln and Guba [23].

#### 2.5.1. Ensuring Credibility

After recording the participant’s speech on the audio recorder, the researchers created a transcript based on the recorded content.

#### 2.5.2. Ensuring Transferability

When recruiting, we assessed the epidemiological background of the participants in terms of age, sex, length of time since transplant, kidney donation types (living or deceased kidney donation), and relationship with the donor. Additionally, study participants were selected from multiple institutions to ensure good generalizability.

#### 2.5.3. Ensuring Dependability

The description and content analysis processes were carried out by the researcher N.M. and were reviewed regularly by co-researchers Ma.M, K.S., A.H., N.M., and M.M. (those had experiences in qualitative research and nurses specializing in chronic diseases) to ensure that there were no differences in the perspectives of the multiple researchers conducting the analysis.

#### 2.5.4. Ensuring Confirmability

In the analysis process, the researcher worked with research supervisor MoM who had experience in qualitative research to ensure that the results extracted were not biased. The results were also checked by nurses with experience in kidney transplants and professionals such as RTCs who ensured there were no lapses in the content analysis. The findings and interpretation were further confirmed by feeding them back to all study participants (member checking).

## 3. Results

### 3.1. Participants’ Attributes and Transplant Backgrounds

Nine participants from three facilities participated in the research. Table 1 summarizes their sociodemographic characteristics. The median age of the participants was 57 (in the range of 36–65) years. There were two males and seven females, and six participants were employed. Eight participants received living kidneys, and one received a donated deceased kidney. Among the living kidney transplants, five were spousal donations and three were parental ones. The median number of months after transplant was 49 (in the range of 13–135). Eight participants had dialysis before the transplant, seven performed hemodialysis, and one peritoneal dialysis. Seven participants had the primary cause of CKD as chronic glomerulonephritis, mainly immunoglobulin A (IgA) nephropathy. The others had diabetic nephropathy, systemic lupus erythematosus, and hypertension.

Participants had CKD stages from 2T to 4T, with 3bT being the most common [24]. During the study period, they were suffering from anemia, obesity, hypertension, and dyslipidemia. One participant was being treated for kidney cancer.

### 3.2. Factors That Maintained and Promoted Self-Management Behaviors

We analyzed the contents, and 115 codes were extracted for the factors resulting in the maintenance and promotion of self-management behaviors. As a result of the classification of each code based on semantic similarity, eight categories were generated, which include twenty-four subcategories. The categories were past painful experiences, attention to change in one’s own body, increased self-efficacy, the autonomy to protect one’s body, the establishment of lifestyle habits, gratitude for the kidney donation, support from family and others, and good partnerships with medical care providers (Table 2).

To explain, the category ‘[ ]’ and the subcategory ‘< >’ is used.

Figure 1 shows the relationships among the thematic categories. To maintain and promote self-management behavior, patients became more attentive to taking care of their bodies as they had past painful experiences before and after kidney transplants. Patients’ sense of self-efficacy was improved through successful self-management experiences, leading to the development of autonomy to protect their bodies, which became an established part of their lifestyle. Gratitude to kidney donors for their immense contribution, obtaining support from family members and others, and developing a good rapport with medical professionals led to the recipient’s greater attention to changes that occur in their body. It fostered a sense of self-efficacy and autonomy to establish a healthy lifestyle. However, as the patient’s disease became stabilized and the duration of expected life expectancy increased, the threat of worsening disease waned, priorities shifted in the patients’ lives, and they became less motivated to control their disease.

[1. Past painful experience].

The fear of worsening kidney function due to past dialysis experience and past experiences with symptom exacerbation promoted the development of patients’ behaviors. Because of their experiences with dialysis before receiving a kidney transplant, some participants had a strong desire not to go through dialysis again, motivated by continuing self-management.

*I was unable to move after dialysis, I couldn’t move my fingers, I couldn’t drink water, and I couldn’t eat anything*.(Participant D)

*I’ve had a sudden dialysis experience, and I’m afraid my kidneys will get worse*.(Participant C)

The individuals’ painful symptoms and the infected experiences of some of the recipients during their care, even before the kidney transplant, led them to take care to maintain their condition. A young recipient woman in her thirties had no child when she was on dialysis. She was determined to have a child and decided to undergo a kidney transplant. She had a history of urinary tract infections associated with her marital life. She discussed her precautions to prevent infections due to conjugal life again, as she would be susceptible to being infected after the transplant by taking immunosuppressive medications.

*I drink water frequently as if I drink too little water, my legs cramp in the morning*.(Participant E)

*After experiencing urinary tract infections due to conjugal life, I feel that I can get infected due to my medication intake and physical condition, and I am trying to be careful*.(Participant E)

[2. Attentiveness to change in one’s own body].

Many recipients received self-monitoring guidance and education from physicians, nurses, and RTCs before and after hospitalization to measure their blood pressure and weight at home after the kidney transplant. They changed their bodies through self-management behavior to improve their physical condition, disease symptoms, and vital sign readings. They also paid attention to their bodies and changed their behaviors when the doctors pointed out that their creatinine levels were high. They reviewed their conditions by observing changes in serum creatinine levels during regular clinic visits and adjusting their daily life activities.

*When my blood pressure goes up, my legs become swollen, I feel my body gets very tired*.(Participant I)

*Then, I take a rest because I think I am moving too much*.(Participant D)

*Because I was told that my creatinine was high, I reduced the amount of salt intake*.(Participant I)

*Visiting the hospital ensures confirmation of what I have done regarding self-management behavior*.(Participant I)

Owing to the susceptibility to infection associated with immunosuppressive medication, one participant became aware of the increased risk of infection. Therefore, they were very careful during the flu season. Moreover, the participants recognized that adequate fluid intake was required in order to maintain kidney function after transplant. They reviewed their water intake and proactively obtained advice from others; they paid attention to their behavior by understanding the actual amount of water they needed to consume.

*I get sore throat easily in the winter, so I gargle and wash my hands several times a day*.(Participant I)

*I prepared a container to measure water intake for the last month or so, and I realized that I was not drinking enough*.(Participant I)

[3. Increased self-efficacy].

Recipients are confronted with the responsibilities associated with personal management in their daily self-management activities. This responsibility involves the pressure not to damage the kidney they received, which led to a variety of stresses, including the fear of not wanting to go through dialysis again, a painful experience in the past. However, amid such burdens, others’ approval of their self-management behaviors was linked to encouraging the recipient’s self-management behavior. The recipient’s isolated feelings were diminished when others acknowledged them and approved of their efforts. In addition, the realization of the effects of behavior was one of the factors in developing self-efficacy for maintaining and promoting self-management behaviors. The effect of diet and medication management behaviors is to keep the patient in good physical and mental health without worsening kidney function.

*I am happy when people recognize that I am doing my best self-management, and it makes me want to do my best again*.(Participant I)

*Transplanting is hard, but if you do self-management behavior, you will feel so much better*.(Participant I)

[4. Autonomy to protect one’s body].

The recipients were aware of the importance of acting autonomously to maintain their physical condition with a transplanted kidney. They recognize that they are much healthier now after kidney transplants than they used to be, and only they can protect their own body from illness, knowing that their condition could worsen again someday. Therefore, they proactively take care of themselves as instructed by the medical staff.

*Doctors only gave me rough instructions for dietary restrictions, but I will be careful to judge the rest on my own*.(Participant I)

[5. Establishment of lifestyle habits].

To maintain the function of the newly acquired kidney as much as possible, there are situations in which the recipient is required to change his or her medical care behavior, which is significantly different from before the kidney transplant. For instance, before the kidney transplant, during the preservation of end-stage renal disease (ESRD), the ability of the kidneys to produce and release urine from the body is significantly reduced, and the amount of water consumed per day is greatly limited and influenced by individual body weight and urine output. However, after kidney transplants, the suggestions have changed, and adults need to drink at least 2 L of water per day to maintain kidney blood flow. The recipient transitioned and acclimated to the new post-transplant routine.

*Once I used to do it, I naturally started writing down the amount of water I took and the time I urinated every day in a notebook*.(Participant B)

On the other hand, some participants acquired this habit in their original lifestyle before transplanting. It influenced the maintenance and promotion of self-management behaviors, even after the transplant. In response to the importance of self-monitoring, measuring, and recording blood pressure and weight, the patient said:

*I keep a pocketbook at home, and I don’t mind writing in the first place*.(Participant G)

[6. Gratitude for kidney donation].

Donors are essential for the success of kidney transplants. Participants expressed their gratitude to the donors and desired to take care of their newly transplanted kidneys. Affection for the donated kidney also influenced recipients to be more proactive in self-management.

*I am grateful to my husband for being a donor of mine, and I want to cherish it*.(Participant A)

*I want to keep my kidney healthy for as long as possible*.(Participant I)

[7. Support from family and others].

Eight of the nine participants were married and all of them were living with their spouses or other relatives. Family members were concerned about their physical condition in daily life. They had opportunities to sit with the participants at the same dining table and cooperated in dietary management. This motivated them to continue their dietary regimes.

*My husband is more concerned about my health than his own*.(Participant I)

*My family is concerned about my health condition, and they eat low-sodium foods, as I do*.(Participant I)

To consider the gratitude of family cooperation, the recipients did not want to cause their family trouble by becoming sick again. They were concerned about the impact of deteriorating health on their family members.

*I did not want to give my mother, the donor, a painful or unpleasant experience if my kidney deteriorated*.(Participant C)

*I am worried that my family will be inconvenienced if I am hospitalized for a urinary tract infection, so I want to avoid that as much as possible*.(Participant A)

Along with family members, participants also received support from their colleagues at work, including time off and benefits. It motivated them to return the favor by practicing self-management after the transplant. Therefore, they wanted to continue their work and contribute to the company. They returned to society, recognizing that the role expectations from others helped them to maintain and promote self-management behavior.

*I want to contribute to my company as the company gave me time off and benefits during my treatment which allowed me to have the surgery without worry*.(Participant D)

Moreover, support from friends with similar illnesses is also an important factor for maintaining and promoting self-management behavior. One participant stated that the transplant improved her health, and she started to take up new hobbies such as Pilates which she did not have before the transplant. Expanding friendships after the transplant provided an opportunity to increase support for the recipients. The desire to respond to support for the family and others was a facilitating factor that assisted in the continuing self-management.

*I can talk with my friends who have experienced the same kidney failure because we can understand each other’s difficulties*.(Participant G)

*I have to know people who are not related to my illness, which broadens my world and refreshes me. I enjoy interacting with people I have met through Pilates, which I started after my transplant surgery. I have broadened my world by getting to know people who have nothing to do with my disease. It is refreshing to laugh and exercise with the friends I have made there. Since I started Pilates, I have not caught a cold at all*.(Participant I)

[8. Good partnership with medical care providers].

Recipients realized that a cooperative good partnership was developed by doctors and RTCs after the transplant. Each time they visited the clinic after the transplant, doctors and RTCs supported them in self-management behaviors. In particular, the participants trusted the RTCs with whom they usually had contact at the clinic. These reliable medical care providers supported the recipients, especially in the mental aspect, and led them to implement regular care as instructed. They are like supporters who are always there cheering and encouraging them. Moreover, the approval of self-management behaviors by medical care providers helped recipients gain confidence in their self-management behaviors, and this had a more sustained and promoting influence. Furthermore, an appropriate explanation for regularly taking immunosuppressant drugs from medical care providers enabled them to acquire the proper knowledge of what they should do.

*I enjoy coming to the clinic because the doctors and coordinators listen to me carefully every time*.(Participant I)

*The coordinators act like supporters who are always there to cheer me up*.(Participant I)

*When the doctor or RTC asks me whether I wash my hands or gargle at home, I feel bad because I haven’t done those things*.(Participant I)

*I’m so happy when the staff around me understand my blood pressure and what I do every day and call me out on it. I feel blessed*.(Participant I)

### 3.3. Factors That Interfered with Self-Management Behaviors

The researcher asked questions about the triggers for why the recipient stopped continuing to perform self-management behaviors and extracted the narrative portions in such a way that it did not interrupt the meaning of the context. Based on the content of the data, 28 codes were extracted for the factors that inhibited self-management behaviors.

Three categories were generated, consisting of seven subcategories (Table 3).

The category names were “[fading threat of worsening disease]”, “[shifting priorities]”, and “[decreased motivation to control the disease]”.

[1. Fading threat of worsening disease].

While one female recipient was aware of the need to continue her daily self-management activities after the transplant, she also experienced a complicated psychological situation. During the interview, she talked about how her physical condition had greatly improved since the transplant. With the increasing post-transplant period, she became less aware of being a kidney disease patient. This seemed to be true for others interviewed, as well.

*I’ve been less aware that I will be sick lately. Every year after transplanting, I feel that I am able to live the same life as others around me, and I am no longer aware that I am sick*.(Participant I)

Several recipients loosened the restrictions suggested by the physicians by making adjustments of their own. When patients noticed there had been no deterioration in either their subjective symptoms or laboratory data, they paid less attention to compliance with doctors’ instructions.

*I believe that the laboratory data has not changed and is in the same condition as when I was fine*.(Participant A)

*I am not strictly restricting salt because the blood test has not gotten worse*.(Participant A)

*I have not felt any symptoms that would make me aware of my deteriorating health condition*.(Participant D)

In addition, based on the experience and knowledge gained during the long period of medical treatment before the kidney transplant, there was confidence in recognizing the deterioration of one’s physical condition.

*I would definitely be able to tell myself if my physical condition worsened*.(Participant G)

[2. Shifting priorities].

After the transplant, recipients are freed from dialysis treatment, which also frees them from time constraints and gives them time to re-engage in social roles at work and home. However, such roles can sometimes be overloaded. There were episodes of unbalanced dietary habits and skipped self-monitoring because of a busy lifestyle. The recipients’ self-management behavior was also affected by the excessive involvement in social activities. After the kidney transplant, their priority was keeping up with their busy schedules, such as work, housework, and hospital visits; therefore, self-management took a backseat. Moreover, the female recipient, who had responsibility for all household chores in her home, understood the importance of having a well-balanced diet with a variety of ingredients, but she could not afford it. Prioritizing daily life was also a factor influencing self-management behavior.

*Skipping blood pressure measurements occasionally because I’m too busy with the household or some other activities*.(Participant E)

*Move according to my daily events, so sometimes medication is not my first choice*.(Participant I)

*I have a family, so when I go shopping, I can’t afford the expensive ingredients and end up eating the same things*.(Participant E)

[3. Decreased motivation to control the disease].

Regarding self-management behaviors that the recipients continued to perform daily after the transplant, there were occasions when their motivation for self-management decreased due to temporary psychological changes. This led to a decrease in their motivation to take care of themselves and affected self-management behaviors. Decreased motivation for medical treatment also affects self-management behaviors.

*Sometimes I don’t measure my blood pressure because I just don’t feel like doing it today*.(Participant E)

*During the day when my family is not around, I sometimes end up eating unbalanced meals because I feel it is too much trouble to cook a proper meal for only me*.(Participant G)

### 3.4. Characteristics of the Extracted Factors According to the Difference of Length of Post-Transplant Timing

Of the three categories of inhibiting factors, codes for <confidence in recognizing deterioration of one’s physical condition>, <lack of awareness of being a kidney disease patient>, and <no deterioration in subjective symptoms> were obtained only from participants who were transplanted for more than five years.

## 4. Discussion

### 4.1. Factors That Maintain and Promote Self-Management Behavior in Post-Kidney Transplant Recipients

The results of this study suggest that post-transplant recipients were influenced by the painful experiences they had before the transplant. Moreover, they became sensitive to the changes that occurred in their bodies, and they acted autonomously to maintain their health. Thus, the fact that such behavior had become habitual before the transplant may have been a factor in the appropriate implementation of self-management behaviors after the transplant. Another influencing factor was the feeling of gratitude for getting support from donors and others, which led to an increased sense of self-efficacy and motivation to continue self-management.

The first factor that promoted the maintenance of self-management behavior after a kidney transplant was [past painful experiences]. Past distress refers to the experience of receiving dialysis therapy during the preservation phase of pre-transplant kidney failure. Most of the participants experienced physical, psychological and social distress when their condition deteriorated and underwent dialysis as dialysis significantly restricted their lives. In our study, seven participants received hemodialysis, and one received peritoneal dialysis before a kidney transplant. Participants required treatment several times a week for hemodialysis and several times a day for peritoneal dialysis. Therefore, patients had many opportunities to experience changes in their bodies, especially blood pressure and weight during their daily self-monitoring [25]. In the previous studies, “aversion to dialysis” was identified as one of the “prevailing fears of consequences”, and negative emotions from past illness experiences reinforced self-management behaviors [26,27] which were consistent with our results. Therefore, the participants continued to engage in appropriate self-management behaviors to avoid painful experiences.

The participants were sensitive to changes in their bodies; thus, they were more attentive to taking care of themselves. Jamieson et al. also found that developing bodily intuition is the key factor that strengthens recipients’ self-management behavior [26]. Some patients gained a new understanding of and trust in their body’s signals [28]. Our results were consistent with this finding that the recipients were highly aware of their perceived body state and sensitive to changes in their bodies.

By taking care of themselves, patients experienced better self-management, which was recognized and praised by others [29], leading to improved self-efficacy. Jamieson et al. stated that gaining confidence in self-management helped patients feel that they were in control of their health [26]. This fostered autonomy or the belief that only he or she could protect his or her body. It can be inferred that the participants recognized that they should be able to understand and deal with any abnormalities that occur in their bodies, and that was important in making autonomous efforts to prevent their health conditions from deteriorating. It has been shown that patients are empowered by independence and incorporate their responsibilities into an automatic routine [30].

The second factor that significantly influenced the maintenance and promotion of self-management was the appreciation of the donors and their surroundings. The [gratitude for kidney donation] category expresses the gratitude that the donor was able to provide a transplant and maintain his or her health. It includes the subcategories <gratitude to the donor for kidney donation> and <affection for the donated kidney>. One woman remarked that she was grateful to her husband for being a donor and wished to maintain a transplanted kidney as long as possible. In Japan, the waiting period for a deceased donated kidney transplant is extremely long (approximately 15 years) because of the shortage of donors, forcing patients to rely on living kidney donors [31]. Recipients are more strongly appreciative of being able to receive a kidney transplant because it is a rare opportunity to be fortunate enough to have a relative eligible as a donor.

It was inferred that our recipients of living donor kidney transplants were more likely to feel gratitude to the donors as they had close contact in their daily lives. The results of the previous study also showed expressions of gratitude as expressed in “debt of gratitude” [15], “indebtedness to the donor” [26], and “enacting a moral duty” [27]. The participants also felt both gratitude and a sense of guilt [32]. Therefore, some felt an obligation not to give up keeping their kidney healthy and not receive complaints of negligence for self-management [27,33]. It has been suggested that patients’ attitudes and beliefs about self-management vary by donor type, age, and medical situation. Patients who have undergone living donor kidney transplants have a strong sense of responsibility for self-management [26,27]. As most (8 out of 9) of our participants received living kidneys, we could not differentiate the self-management behavior among them.

In our study, patients’ greater appreciation for their families was largely because they lived with family members who were donors and further helped them with self-management in their lives. There is also gratitude to colleagues who helped, along with family members. Another strength of this study is that participants had the opportunity to talk about themselves with others who had the same disease, as they could understand their struggles. As peer learners, recipients received advice based on others’ experiences learning about kidney rejection and health monitoring [6]. A recipient said that, after the transplant, he felt refreshed, and his world expanded as he was able to connect with people unrelated to his disease. In the literature, a patient with kidney failure said that dialysis had narrowed his world and made it very restricted [25,27], which led to feelings of confinement and helplessness, affecting his ability to participate in social activities he previously enjoyed [34,35], and making it difficult to maintain social connections and friendships [19]. The recipients experienced a new, joyful world after the transplant.

Moreover, a good partnership with medical care providers also facilitates self-management behavior. Transplant teams were viewed as “saviors” [36] who helped patients resume normal life [28,33]. Recipients maintain an adherence to self-management as a way of expressing gratitude [34]. Achieving self-efficacy and having a sense of accountability to their donors, medical teams, and transplant recipient peers strengthened the recipients to maintain an adherence to their self-management tasks [6,26]. The presence of social support, including verbal persuasion from others, has also been found to increase patient self-efficacy [37] and is an influencing factor in maintaining and improving self-management.

### 4.2. Factors Inhibiting Self-Management Behavior in Post-Transplant Recipients

The influencing factors that inhibit self-management behavior as the post-transplant course stabilized the threat of worsening disease faded out, life priorities shifted, and a decrease in motivation to control the disease state occurred, leading to an overall decrease in the implementation of self-management.

Our results indicate that the fading threat is caused by the absence of worsening laboratory data and subjective symptoms and by a diminished awareness of kidney disease. The first or the second year post-transplant tended to be a period of thorough self-management for the adjustment of a new post-transplant life, and many tasks were involved, including medical management and social and emotional control [18,29]. After the kidney transplant, patients are obsessed with the constant awareness of having a transplanted kidney [38], and their thoughts are dominated by avoiding transplant failure [32] and protecting the transplanted kidney [26]. However, some perceived themselves as healthy, as being cured of chronic kidney disease and normal, and as having no acute health problems [39]. Others wanted not to feel like patients [36], and some denied being patients and stopped hospital follow-up appointments and medications, as they began to think they became normal, and everything was fine [40].

Over time, some suffered from burnout and found taking medications and attending medical appointments to be tedious and stressful [26]. It was also inferred that, beyond the period of instability following the transplant, as patients became accustomed to and adapted to the changes, they developed a sense of control. The threat that they felt at the beginning of the transplant such as the loss of a transplanted kidney may have diminished, and appropriate self-care behaviors may become less thorough. Only participants with a more-than-five-years-long post-transplant follow-up period inhibited their self-management, i.e., <confidence in recognizing deterioration of one’s physical condition>, <lack of awareness of being a kidney disease patient>, and <no deterioration in subjective symptoms>. When we analyzed the backgrounds of the recipients, three (A, G, and I) of them said the code included <confidence in noticing one’s physical condition deteriorating>, who had transplantation for more than 5 years.

A study on non-adherence to immunosuppressive medication among post-transplant recipients observed that non-adherence increased with the time that had elapsed after transplant [19]. It is thought that a prolonged period after a transplant reduced the sense of threat of the disease which might affect self-management behaviors. Kidney transplant recipients develop allograft failure over time, requiring dialysis initiation or re-initiation [41]. Therefore, self-management is imperative in preventing renal allograft failure. It has been reported that 60% of post-kidney transplant patients have CKD stage 3 or higher, so even those who are less aware of the need for continued self-management after a long period post-transplant should be aware of this [42]. To compare the three categories of inhibiting factors, codes for <lack of awareness of being a kidney disease> and <no deterioration in subjective symptoms> in <confidence in recognizing deterioration of one’s physical condition> were only observed for the participants who had been post-transplant for ≥five years compared to those who were less than five years post-transplant. It is not easy to maintain a high level of self-management behaviors for a long period after a transplant [19]. Therefore, it is important to create educational opportunities for medical care providers to recognize the “fading threat of worsening disease” in the recipient, and to set short-term and long-term behavioral goals together to determine what needs to be done to maintain appropriate self-management behaviors.

After the transplant, the patient no longer had ESRD and did not require dialysis, strict nutrition, medication, and fluid management. This change provided an opportunity to regain social roles, it was thought that there would instead be a decrease in the priority of self-management behaviors and reduced motivation to control the disease. Regarding shifting priorities, kidney transplants provided a major change in the treatment environment of the recipients, which allowed them to regain time to work and carry out household activities. They have the least restrictions on work capacity compared to dialysis and CKD preservation patients, with more than 91% being able to continue working in the long term [43]. Kidney transplant recipients can selectively perform a variety of tasks depending on the value they place and considering the possible side effects and complications, and how they prioritize them [36]. A diminished threat to disease management and change in priorities over time after a transplant decrease motivation to control the disease.

### 4.3. Implementation of Study Findings in the Clinical Field/Clinical Implications of Study Findings

The results of this study suggest that there is a need to create a support system including medical care professionals for the management of post-kidney transplant patients. Over time after a kidney transplant, the threat of worsening disease fades, and self-management turns self-directed. This has the potential to cause a reduction in kidney function and exacerbate the effects of comorbidities. Therefore, medical care providers need to follow up with kidney transplant patients after five years to ensure that they can manage their health effectively. Providing adequate knowledge and motivating them through interviews with medical care providers can help counter the reduced awareness over time (the “fading”) of threats to medical care, and prevent long-term complications and problems.

In many cases, transplant patients are referred to their family physicians for long-term care. In general, nurses in primary care lack knowledge about self-management skills for post-transplant patients; a transplant team can develop a transitional care system to collaborate, educate, and support primary care physicians [12]. In Japan, because of the lack of transplant opportunities, short-term post-transplant management is performed by physicians and nurses at specialized hospitals. Therefore, it is necessary to create educational kits for nurses and to conduct educational awareness activities.

### 4.4. Study Limitations

This study has certain limitations. We did not include the participants aged < 20 years; this might cause the exclusion of a significant proportion of the younger-age kidney transplant population who might have unique challenges and experiences related to self-management. The sample size of nine participants is relatively small, which limits the generalizability of the study findings. We enrolled participants from different hospitals, as, in Japan, it has been noted that the content of self-management guidance for kidney recipients differs from facility to facility [44], and we did not compare the influencing factors of different hospitals, which might influence the results of this study. This study was based on limited narratives of the participants who agreed to participate. Therefore, the results of this study are limited by the possibility of bias in the participants’ transplant environments. In the future, it would be desirable to undertake a larger qualitative study that replicates the methodology used here in order to confirm the picture emerging from this exploratory investigation. This could then be supplemented with a quantitative multi-hospital study to confirm the general applicability of the identified factors and their prevalence in the population of transplant patients. 

## 5. Conclusions

This study provides valuable insights into the factors influencing self-management behaviors in kidney transplant recipients. We conducted semi-structured interviews to identify factors influencing the self-management behaviors of recipients who had more than one year after a kidney transplant. We extracted eight categories that maintained and promoted self-management behavior: [attentiveness to changes in one’s own body], [good partnership with medical care providers], [past painful experiences], [establishment of lifestyle habits], [autonomy to protect one’s own body], [support from family and others], [gratitude for kidney donation], and [increased self-efficacy]. We also extracted three categories that inhibited self-management behavior: [fading threat of worsening disease], [shifting priorities], and [decreased motivation to control the disease]. The passage of time after the transplant became a barrier to continued self-management because the threat of disease deterioration diminished, and the performance of work and household chores became a higher priority than self-management implementation. Therefore, we believe it is necessary to develop a transition care system in which nurses interview patients whose data show a trend toward deterioration over time after the transplant, providing them with the knowledge of correct self-management methods to monitor their disease. The identified factors can be addressed for the development of targeted interventions and support programs to help recipients maintain optimal health and improve long-term outcomes.

## Figures and Tables

**Figure 1 healthcare-12-02264-f001:**
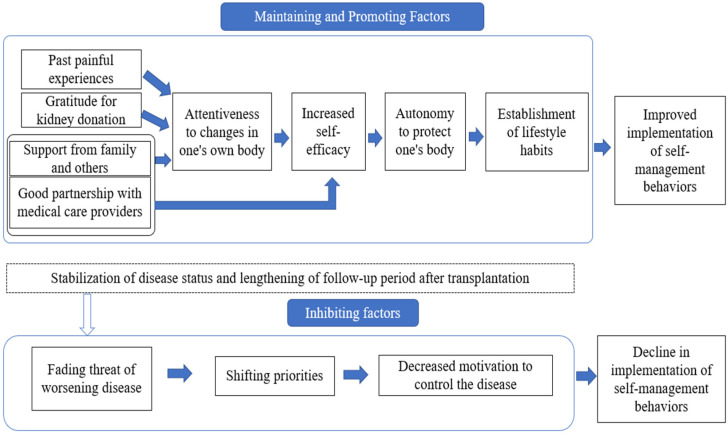
Factors that promote and inhibit self-management behavior.

**Table 1 healthcare-12-02264-t001:** Participants’ characteristics and transplant background.

Participants	Participants Characteristics				Matters Related to Kidney Transplant							Current Medical Treatment Status	
	Age (Years)	Sex *	Employment	Marital Status and Living Alone	Types of Transplants	Relation Ship to Donor	Donor’s Current Physical Condition **	Relationships with Donors **	Post-Transplant Elapsed Time (in Months)	Pre-Transplant Dialysis Types ***	Primary Cause of Kidney Disease	CKD Stages (T)	Complications of Illness
A	36	F	Yes	Married No	Living kidney	Spouse	Very good	Fairly good	65	HD	Glomerulonephritis	2	Anemia
B	36	F	Yes	Unmarried No	Living kidney	Parent	Fairly good	Very good	29	HD	Glomerulonephritis	G3b	Obesity
C	38	M	Yes	Married No	Living kidney	Parent	Very good	Very good	49	HD	Systemic lupus erythematosus	3b	Hypertension
D	65	M	Yes	Married No	Living kidney	Spouse	Very good	Very good	14	HD	Hypertension	4	Hypertension Dyslipidemia
E	57	F	No	Married No	Living kidney	Spouse	Fairly good	Fairly good	43	None	Diabetes	2	Diabetes Cataract
F	61	F	No	Married No	Living kidney	Spouse	Slightly bad	Fairly good	13	HD	Glomerulonephritis	3b	Asthma Kidney cancer
G	60	F	Yes	Married No	Deceased kidney				135	HD	Glomerulonephritis	3b	Hypertension Hypothyroidism
H	57	F	No	Married No	Living kidney	Parent	Very good	Very good	78	PD	Glomerulonephritis	3b	Dyslipidemia
I	51	F	Yes	Married No	Living kidney	Spouse	Very good	Very good	68	HD	Glomerulonephritis	3a	None

* F = Female, M = Male. ** At the interview, the recipient’s level of awareness of the items was rated on a 5-point scale (Very good, Fairly good, Neither, Slightly bad, Very bad). *** HD = Hemodialysis, PD = Peritoneal dialysis.

**Table 2 healthcare-12-02264-t002:** Factors that maintain or promote self-management behaviors.

Categories	Subcategories	Statements	Number of Codes	Total Number of Codes
Past painful experiences	Fear of worsening kidney function due to experience of symptom exacerbation	After experiencing urinary tract infections during my marital life, I am concerned about the possibility of reinfection based on my medication and physical condition, so I am making an effort to be careful. If I drink too little water, I get leg cramps in the morning, so I drink water frequently.	9	18
	Fear of worsening kidney function due to experience of dialysis	I didn’t want to be on dialysis further as it was difficult for me to move after dialysis was over. My fingers were cramped, I couldn’t drink water, and I was restricted in what I could eat. I don’t want to be on dialysis again because I have experienced sudden dialysis, and I am afraid that my kidneys will deteriorate.	9	
Attentiveness to changes in one’s own body	Recognize changes in physical condition	My blood pressure was higher in the winter, so I consulted with my doctor. When my blood pressure rises, my legs swell, or I feel very tired, I think I’m moving too much and should rest.	8	24
	Notice changes in serum creatinine level	When I see my creatinine is elevated, I know I have to be very careful. Very hard. I’m trying to cut off my salt intake as the doctor said my creatinine is high.	4	
	Review through regular outpatient visits	When my doctor’s appointment is coming up, I worry about drinking water and ensuring my blood pressure doesn’t rise. Coming to the hospital is a reconfirmation of what I’ve been doing.	5	
	Awareness of the increasing risk of infection	During the flu season, I’m very careful about infection. I get a sore throat easily in the winter, so I gargle and wash my hands several times a day.	4	
	Understand the actual amount of water consumption	I prepared a container to measure water one month before, and I think I didn’t do enough.	3	
Increased self-efficacy	Others’ approval of self-management behaviors	I have understanding and support for my efforts. I am happy when I am recognized for my efforts in self-management, and I try to do it again.	2	3
	Realization of the effects of self-management behaviors	Transplanting is hard work, but I realize that if I do it right, I’ll feel so much better.	1	
Autonomy to protect one’s body	Consciousness of proactive health management	Only I can protect my body from disease. The doctor will only give me rough guidelines for dietary restrictions, but I’ll be careful to judge the rest for myself.	9	12
Establishment of lifestyle habits	Habitualization of self-management behavior acquired after transplant	Once I got used to it, I naturally started to write down the amount of water I took in and the time of urination every day in a notebook. It has become a habit for me to put a scale in front of me, measure the amount of water I drink, and write it on a piece of paper.	10	15
	Lifestyle habits acquired before the transplant	I also keep a pocketbook at home, and I don’t mind writing in the first place. I like water, so I drink a lot.	5	
Gratitude for kidney donation	Gratitude to the donor for kidney donation	I am grateful that my husband became a donor, and I want to cherish that.	7	11
	Affection for the donated kidney	I want to keep my current kidney healthy for as long as possible.	4	
Support from family and others	Recognize others’ role expectations	I want to contribute to the company because the company gave me time off and benefits during my treatment, so I was able to have the surgery without worry. I want to continue working to protect my family.	3	19
	Family concerns about physical condition	My husband cares more about my health than his own. My wife tells me I’m taking too many calories, so I need to be careful.	7	
	Cooperation of family members in diet management	My family cares about my health and eats low-sodium-containing foods as well.	2	
	Support from friends with similar diseases	With friends who have undergone the same kidney failure, we can understand each other’s struggles and talk about our own.	2	
	Concerns about the impact of deteriorating health on family members	I don’t want my family, or mother who is a donor, to feel pain or uncomfortable because of my kidney failure. I am worried that I will make trouble for my family if I am hospitalized for a urinary tract infection, so I want to avoid that as much as possible.	4	
	Expanded friendships after the transplant	I get to know the people who have nothing to do with my disease, which opens my world and refreshes me.	1	
Good partnership with medical care providers	Presence of reliable medical care providers	The doctors and coordinators listen to me carefully every time I come to the outpatient clinic, and I enjoy coming to the clinic. The coordinator is like a supporter who consistently provides me with support and encouragement.	10	20
	Regular care instructions for my physical condition from reliable medical care providers	When asked if I’m washing my hands or gargling, I often realize, ‘Oops. I haven’t.’ When asked how I manage myself daily, I think I should talk about it.	7	
	Appropriate explanation of an immunosuppressant from medical care providers	At the time of the transplant, I was properly informed in the ward about the effects of immunosuppressive drugs and what to do if I forgot to take them.	2	
	Approval of self-management behaviors from reliable medical care providers	I’m so happy when the staff around me understand my blood pressure and what I do every day and call me out on it. I feel blessed.	1	
Total	24 subcategories			115

**Table 3 healthcare-12-02264-t003:** Factors that inhibit self-management behavior.

Categories	Subcategories	Statements	Number of Codes	Total Number of Codes
Fading threat of worsening disease	Lack of awareness of being a kidney disease patient	Lately, I’ve been feeling less and less aware of my illness. Every year after the transplant, I feel that I am able to live the same life as the people around me, and I am no longer aware that I am sick.	6	14
	No deterioration in laboratory data	My lab data hasn’t changed, and I’m assuming I’m in the same state as when I was fine. The blood tests are not getting worse, so I’m not strictly restricting salt.	4	
	No deterioration in subjective symptoms	I haven’t felt any symptoms that make me aware of my condition worsening. I don’t feel swollen, and there’s nothing wrong with me, so I think I’m fine.	2	
	Confidence in recognizing the deterioration of one’s own physical condition	I think I can definitely tell when my health is deteriorating.	2	
Shifting priorities	Excessive increase in social roles	When I’m working a lot, I eat lunch at a convenience store. Sometimes, I skip blood pressure measurements because I’m too busy with the house or something.	5	10
	Prioritizing daily life	Sometimes, medication is not my first choice because I move according to my daily schedule. I have a family, so, when I go shopping, I can’t afford to buy expensive food items and end up buying only the same things.	5	
Decreased motivation to control the disease	Decreased motivation for medical treatment behaviors	Sometimes, I just don’t feel like taking my blood pressure today, so I don’t take it. It’s a hassle to make a proper meal for one person.	4	4
Total	7 subcategories			28

## Data Availability

The data presented in this study are available upon request from the corresponding author. The data are not publicly available due to privacy restrictions.

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
