# Peer review of "Factors Influencing Self-Management Behaviors Among Patients with Post-Kidney Transplantation: A Qualitative Study of the Chronic Phase Transition"

_healthcare, 2024, doi:10.3390/healthcare12222264_

Round 1
Reviewer 1 Report
Comments and Suggestions for Authors
I believe this paper covers a topic of importance to transplant nephrologists. However, I doubt that a study with only 9 participants and without any quantitative analysis whatsoever is suitable to draw the conclusions the authors are drawing. This paper would be significantly improved by an increased number of participants as well as such an analysis.
Comments on the Quality of English LanguageI believe this paper would profit from English language editing by a native speaker.
Author Response
Thank you very much for your valuable comments and further consideration of our manuscript for possible acceptance.
Please find below the point-by-point responses and all the changes are highlighted and marked by “RED” within the revised manuscript.
Best regards
KATM Ehsanul Huq, MBBS, DTM, MSc, PhD
Graduate School of Biomedical and Health Sciences
Hiroshima University, Japan
Mobile:080 6266 8578
Comments: I doubt that a study with only 9 participants and without any quantitative analysis whatsoever is suitable to draw the conclusions the authors are drawing. This paper would be significantly improved by an increased number of participants as well as such an analysis.
Responses: As we conducted this study as a qualitative design, we did not calculate and prefix the sample size. We enrolled the participants up to the ‘saturation’ point when we achieved the study objectives and reached the saturation point and we found there were no new factors determined to be extracted. We revised the statement and mentioned it on page: 3, line:108-112.
We included our sample size limitation in the ‘Study Limitations’ section (page: 17, Line: 631-632, and 637-640).
Comments: I believe this paper would profit from English language editing by a native speaker.
Responses: To improve the English language throughout the paper, all the authors worked on it. We also edited our manuscript by a native English-speaking colleague.
Reviewer 2 Report
Comments and Suggestions for Authors The followings are my recommendations on your essay, I hope it will be useful:1-I recommend to revise keywords.
2-The most common phrase for renal failure is "end stage renal disease; or ESRD .I recommend to use this acronym
3- The most accepted CKD classification is recommended by KDIGO, so I recommend to use this classification, the 3T or 4 T is not familiar for the audiences.
4-I think that nine patients are not sufficient for analysis , please provide evidence of statistical analysis for this sample size.
5-Some words in Table 1 are displaced. e.g. "Relations hip to donor". Please correct them.
6-The discussion should be included by most recent literature review . Comments on the Quality of English Language
Minor editing of English language required.
Author Response
Thank you very much for your valuable comments and further consideration of our manuscript for possible acceptance.
Please find below the point-by-point responses and all the changes are highlighted and marked by “RED” within the revised manuscript.
Best regards
KATM Ehsanul Huq, MBBS, DTM, MSc, PhD
Graduate School of Biomedical and Health Sciences
Hiroshima University, Japan
Mobile:080 6266 8578
Comments 1: I recommend to revise keywords.
Responses: According to your suggestion we revised the keywords.
Comments 2: The most common phrase for renal failure is "end stage renal disease; or ESRD .I recommend to use this acronym
Responses: We corrected it on page 10, line 323 and page 16, line 598.
Comments 3: The most accepted CKD classification is recommended by KDIGO, so I recommend to use this classification, the 3T or 4 T is not familiar for the audiences.
Responses: As we enrolled post-transplant kidney patients, for that we used ‘T’ with the CKD stages. The KDIGO guidelines also recommended that post-transplantation CKD should be assigned the stage ‘T’. After transplantation, the GFR is classified into 1 to 5, the same as the standard for CKD stage classification. The standards are also the same according to KDIGO Transplant guideline 2009: https://kdigo.org/guidelines/transplant-recipient/
Comments 4: I think that nine patients are not sufficient for analysis, please provide evidence of statistical analysis for this sample size.
Responses: As we conducted this study as a qualitative design, we did not calculate and prefix the sample size. We enrolled the participants up to the ‘saturation’ point when we achieved the study objectives and reached the saturation point and we found there were no new factors determined to be extracted. We revised the statement and mentioned it on page: 3, line:108-112.
We included our sample size limitation in the ‘Study Limitations’ section (page: 17, Line: 631-632, and 637-640).
Comments 5: Some words in Table 1 are displaced. e.g. "Relations hip to donor". Please correct them.
Responses: We corrected those accordingly in Table 1.
Comments 6: The discussion should be included by most recent literature review.
Responses: We included the updated references in the ‘Discussion’ part (References number: 2, 5-9, 13, 25, 28, 30 and 42).
Comments: Minor editing of English language required.
Responses: To improve the English language throughout the paper, all the authors worked on it. We also edited our manuscript by a native English-speaking colleague.
Reviewer 3 Report
Comments and Suggestions for Authors
Dear authors,
I have made a great historical analysis. The main question, I am wondering is, are you really sure that since 10 years ago the behavior hasn't been changing?
Overall Assessment
The study presents valuable insights into the factors influencing self-management behaviors in kidney transplant recipients. The qualitative methodology allows for a deep exploration of the complex factors at play. However, there are a few areas that could be strengthened to enhance the article's impact.
Positive Aspects
- Relevance: The topic is highly relevant given the importance of self-management in post-transplant care.
- Depth of Analysis: The qualitative approach allowed for a detailed exploration of factors promoting and inhibiting self-management.
- Identification of Key Factors: The study identified several important factors that can influence recipient behavior, providing valuable insights for healthcare providers.
Areas to be Improved, IMHO, are the following:
- Timeliness of Data: The data collection period (April-December 2016) is relatively old. While the principles of self-management may not have changed significantly, it would be beneficial to update the data to reflect more recent trends and practices in post-transplant care.
- Age Restriction: The age limit of 20 years might exclude a significant portion of the kidney transplant population, particularly younger individuals who may have unique challenges and experiences related to self-management.
- Generalizability: The sample size of nine participants is relatively small, which limits the generalizability of the findings. It would be valuable to consider replicating the study with a larger sample to confirm the identified factors and their prevalence.
My Conclusion
Despite these limitations, the study provides valuable insights into the factors influencing self-management behaviors in kidney transplant recipients. The identified factors, I hope, can inform the development of targeted interventions and support programs to help recipients maintain optimal health and improve long-term outcomes.
Author Response
Thank you very much for your valuable comments and further consideration of our manuscript for possible acceptance.
Please find below the point-by-point responses and all the changes are highlighted and marked by “RED” within the revised manuscript.
Best regards
KATM Ehsanul Huq, MBBS, DTM, MSc, PhD
Graduate School of Biomedical and Health Sciences
Hiroshima University, Japan
Mobile:080 6266 8578
Comments: I am wondering is, are you really sure that since 10 years ago the behavior hasn't been changing?
Timeliness of Data: The data collection period (April-December 2016) is relatively old. While the principles of self-management may not have changed significantly, it would be beneficial to update the data to reflect more recent trends and practices in post-transplant care.
Responses: According to your advice, we added some references in the ‘Introduction’ part those support that self-management behavior is still important (page: 1, line: 41-44).
We also included the updated references in the ‘Discussion’ part.
Comments: Age Restriction: The age limit of 20 years might exclude a significant portion of the kidney transplant population, particularly younger individuals who may have unique challenges and experiences related to self-management.
Responses: You are absolutely right. We included your statement in the ‘Study Limitations’ section (page 17, line 628-631).
Comments: Generalizability: The sample size of nine participants is relatively small, which limits the generalizability of the findings. It would be valuable to consider replicating the study with a larger sample to confirm the identified factors and their prevalence.
Responses: We incorporated your statement in the ‘Study Limitations’ section (page 17, line 631-632 & 637-640).
Comments: My Conclusion
Despite these limitations, the study provides valuable insights into the factors influencing self-management behaviors in kidney transplant recipients. The identified factors, I hope, can inform the development of targeted interventions and support programs to help recipients maintain optimal health and improve long-term outcomes.
Responses: We included your conclusion in the ‘Conclusion’ part on page 17.
Round 2
Reviewer 1 Report
Comments and Suggestions for Authors
I believe you have improved the overall quality of the manuscript, especially regarding the English language. However, my main point of critique was the low number of participants and this point obviously still stands. While I understand that it is impossible to improve upon that years after the initial data collection, I still don't believe this manuscript holds enough scientific merit to be published in this journal.
Comments on the Quality of English LanguageLanguage was substantially improved.
Author Response
Date: 11 November, 2024
Subject: Re-submission of the Manuscript ID healthcare-3228811
Thank you very much again for your valuable comments and further consideration of our manuscript for possible acceptance.
Please find below the point-by-point responses and all the changes are highlighted and marked by “RED” within the revised manuscript.
Best regards
KATM Ehsanul Huq, MBBS, DTM, MSc, PhD
Graduate School of Biomedical and Health Sciences
Hiroshima University, Japan
Mobile:080 6266 8578
Responses:
Query: My main point of critique was the low number of participants and this point obviously still stands. While I understand that it is impossible to improve upon that years after the initial data collection, I still don't believe this manuscript holds enough scientific merit to be published in this journal.
Response: Thank you very much for your observation. You are absolutely right that we did not have sufficient sample size. As a qualitative research, we tried to increase the sample size up to the saturation point. However, there are many limitations in our study, we assume that the findings of our study are probably sufficiently interesting and will provide some scientific evidence for physicians and kidney transplant patients.
Reviewer 3 Report
Comments and Suggestions for Authors
Dear authors, I accept your response.
Author Response
Date: 11 November, 2024
Subject: Re-submission of the Manuscript ID healthcare-3228811
Thank you very much again for your positive comments and consideration of our manuscript for possible publication.
Best regards
KATM Ehsanul Huq, MBBS, DTM, MSc, PhD
Graduate School of Biomedical and Health Sciences
Hiroshima University, Japan
Mobile:080 6266 8578